# TRANSPLANT OF PERCEPTRONS

## ABSTRACT

We propose to *transplant active cells into inactive cells* in neural networks, inspired by the concept of "transplant" in the field of neuroscience, where dead neurons are replaced with live ones to improve brain functions. This is motivated by the fact that a number of major machine learning methodologies such as the perceptron and convolutional neural networks have been invented via the collaboration between neurobiology and computer science. We theoretically discuss how transplant improves the quality of representation of perceptron layers in terms of the mutual information and the loss function with respect to the performance of the whole network. Moreover, we empirically evaluate the effectiveness of transplant in the task of supervised classification. Our proposal is simple and applicable to any neural networks which contain at least one perceptron layer.

## 1 INTRODUCTION

The history of neural networks stretches back to the middle of 20th century, when Frank Rosenblatt proposed the idea of the "Perceptron" in 1958 (Rosenblatt, 1958). The perceptron is inspired by the "formalized neuron", which is the first mathematical model of neural networks presented by Warren McCulloch and Walter Pitts in 1943 (McCulloch & Pitts, 1943). The origin of the mathematical approach to neurons can be traced back to Norbert Wiener, who is the father of Cybernetics (Wiener, 1948) that tries to find common laws of the control and communication between different fields like physics, biology, psychology, or social sciences. Since the perceptron has appeared, neural networks have been evolved with a number of milestone discoveries including convolutional neural networks (Fukushima, 1980; LeCun et al., 1998), backpropagation (Rumelhart et al., 1986), and the attention mechanism (Vaswani et al., 2017). Several pioneers found fundamental technologies while trying to find common rules between neural networks and biological neurons (Churchland & Sejnowski, 1988; Hinton et al., 1984; Hopfield, 1982; Turing, 1950). Although essential studies were performed via collaborations with neuroscience, recently such an interaction has decreased, due to the enormous and complicated growth of both topics. Therefore, looking back at the fusion of those disciplines has been discussed and re-evaluated again (Hassabis et al., 2017).

In neuroscience, the ability of the mammalian brain to recover for neuronal loss caused by disease or injury is hardly limited (Falkner et al., 2016). However, recent studies show that the *transplantation* of neuronal cells (e.g. fatal neurons) into lost cells recover and improve the ability of the brain under some conditions (Grade & Götz, 2017). Moreover, repair of the traumatically injured brain based on the transplantation of neuronal cells to improve memory precision has also been presented (Zhu et al., 2019; Götz & Bocchi, 2021) (Figure 1).

In this paper, we propose the concept of "transplant" in the perceptron, inspired by the above recent advance of transplant techniques in neuroscience. The "activeness" of each cell in the perceptron, which indicates the significance of the corresponding cell, is defined based on the Hebbian learning rule (Hebb, 1949), one of the major theories in neuroscience which represents the law of synaptic plasticity in the brain. To increase the ratio of important cells for active information propagation, we copy active cells into less active cells. We call this operation "transplant" of the cells, as it implants active cells with flexible outputs instead of inactive cells, like grafting embryonic neurons into damaged part of the brain. Transplant is flexible and scalable, since this method is applicable for any neural architectures which contain perceptron layers.

The contributions of this paper can be summarized as follows:

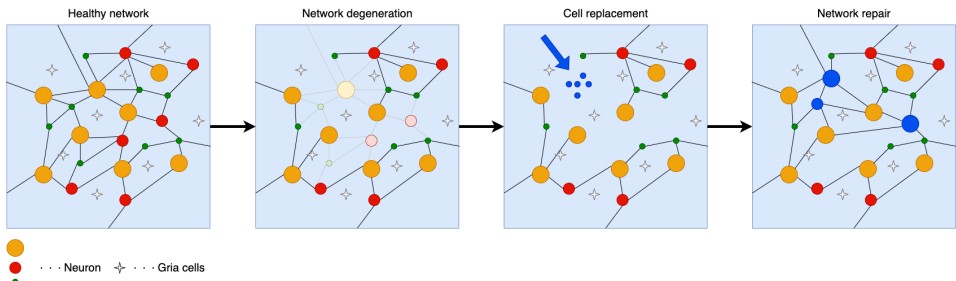

Figure 1: Illustration of neuronal replacement therapy.

---

**Algorithm 1** Transplant

---

**Require:** Percentage of transplant $\eta(\%)$, perceptron weights $\boldsymbol{W} \in \mathbb{R}^{m \times n}$, biases $\boldsymbol{b} \in \mathbb{R}^n$, number of iterations $k \in \mathbb{N}$, concatenated perceptron outputs between checkpoints $\boldsymbol{D} \in \mathbb{R}^{k\beta \times n}$
**Ensure:** $\boldsymbol{W}, \boldsymbol{b}$
  Initialize $\boldsymbol{a}$ as an $n$-dimensional vector $\boldsymbol{a} = a(\boldsymbol{D})$, from equation 1;
  $\boldsymbol{W}' \leftarrow \boldsymbol{W} = (\boldsymbol{w}_1, \boldsymbol{w}_2, \ldots, \boldsymbol{w}_n)$ and $\boldsymbol{b}' \leftarrow \boldsymbol{b} = (b_1, b_2, \ldots, b_n)$;
  Sort $\boldsymbol{W}'$ and $\boldsymbol{b}'$ in descending order of $\boldsymbol{a}$;
  $\boldsymbol{w}_{n-l+1} \leftarrow \boldsymbol{w}'_l$ and $b_{n-l+1} \leftarrow b'_l$ for each $l \in \{1, \ldots, \lfloor (\eta/100)n \rfloor\}$;

---

- We bring the concept of transplant into machine learning, by crossing the fields of neuroscience and neural networks.

- We theoretically analyze the behavior of transplantation in terms of the mutual information.

- We apply our method to supervised training for classification and evaluate it on real-world datasets including the MNIST dataset (LeCun et al., 1998). We show that transplant improves the accuracy for different architectures of the multi-layer perceptron (MLP).

## 2 FORMULATION OF TRANSPLANT

We formulate the operation of transplant and discuss the relationship with neuroscience.

### 2.1 ALGORITHM OF TRANSPLANT

The transplant procedure is formally defined as follows: For each checkpoint, we compute the activeness of each cell in a perceptron, followed by copying (transplanting) $\eta\%$ of cells with higher activeness into the same number of inactive cells with lower activeness. The outline of the transplantation process is shown in Figure 2 and the algorithm is shown in Algorithm 1. Once we define the *activeness* of each cell, the transplant can be performed on any neural architectures, and we propose to use the variance of output values as the activeness. An overview of the process of calculating activeness is shown in Figure 3.

More precisely, given a perceptron with a weight matrix $\boldsymbol{W} \in \mathbb{R}^{m \times n}$ for $m$ dimensional input and $n$ dimensional output and biases $\boldsymbol{b} \in \mathbb{R}^m$. For an input vector $\boldsymbol{x} \in \mathbb{R}^m$, the output $\boldsymbol{y} \in \mathbb{R}^n$ of the perceptron is defined as $\boldsymbol{y} = \boldsymbol{x}\boldsymbol{W} + \boldsymbol{b}$. While training with the batch size $\beta \in \mathbb{N}$, during $k \in \mathbb{N}$ iterations between each checkpoint, we store batches of perceptron outputs and concatenate them as $\boldsymbol{D} \in \mathbb{R}^{k\beta \times n}$. For each column vector $\boldsymbol{d}$ of $\boldsymbol{D}$, we define its activeness $a(\boldsymbol{d})$ as its variance, that is,

$$a(\boldsymbol{d}) := V[\boldsymbol{d}] = E[(\boldsymbol{d} - E[\boldsymbol{d}])^2], \quad \text{where} \quad E[\boldsymbol{d}] = E[(d_1, \ldots, d_{k\beta})^T] = \frac{1}{k\beta} \sum_{l=1}^{k\beta} d_l. \quad (1)$$

In 1949, Donald Hebb proposed the theory "Hebbian learning rule" (Hebb, 1949), which says that if the axon of a cell A is close enough to stimulate another cell B, or repeatedly participates in its firing, a growth process or metabolic change takes place in one or both cells, so that the efficiency of A as one of the cells firing B is increased. In short, "neurons that fire together, wire together".

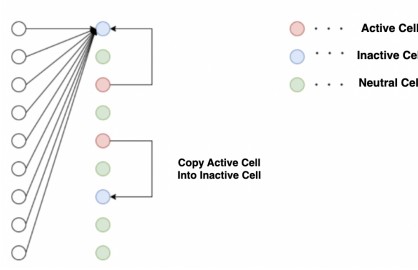

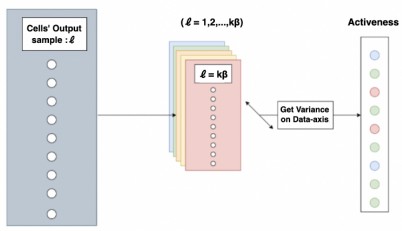

Figure 2: Transplant process in a perceptron.          Figure 3: Activeness.

In the Hebbian learning, if data has zero-mean, the weight vector will ultimately align itself with the direction of greatest variance in the data, and hebbian learning adjust the weight vector so as to maximize the variance in the output (Hebb, 1949). In the architecture of the perceptron, we can interpret the variance of an input cell as the efficiency of A to fire B, and the deviation of signals means the fire for B, since the behavior of the output cell Y is determined by the linear connection of the connected cells X in the neuron of the next layer. Also, there are multiple works that evaluates the importance of cells by measuring the variance of neural response activities (Churchland et al., 2011; Waschke et al., 2021). Therefore, we use the variance of the signals of the perceptron cells to evaluate the activeness of each cell.

## 2.2 Memory-efficient way to calculate the Activeness

In the operation of the transplantation, we store batches of perceptron outputs and concatenate them as $D \in \mathbb{R}^{k\beta \times n}$. In this case, the space complexity becomes $O(k\beta n)$, and more time/memory computational resources are required. However, the equation $V[d] = E[d^2] - E[d]^2$ shows that we do not need to save all of the outputs, but just need to save the sum of the square of each $n$ values and the sum of each $n$ values. We get $E[d^2]$ and $E[d]$ at the checkpoint, and we are able to calculate the activeness from the above equation, which reduces the time complexity to $O(\beta + n)$.

## 3 Theoretical Analysis

In this section, we theoretically analyze the behavior of the perceptron under transplant using the mutual information as an evaluation metric. Moreover, we estimate the impact of transplant with respect to the performance of the model, and explain the statistical functionality of transplant in minimization of the error of neural networks.

Suppose that each input $x_i$ to the $i$-th cell ($i \in \{1, 2, \ldots, m\}$) follows a Gaussian distribution $x_i \sim \mathcal{N}(\mu_{X_i}, \sigma_{X_i}^2)$ with the mean $\mu_{Xi}$ and the standard deviation $\sigma_{Xi}$. From the definition of the perceptron, output $y_j$ of the $j$-th cell ($j \in \{1, 2, \ldots, n\}$) of the next layer is given as

$$y_j = \sum_i x_i w_{i,j} + b_j. \tag{2}$$

From Equation 2 and the reproductive property of the Gaussian distribution, $y_j$ also follows the Gaussian, and its mean $\mu_{Yj}$ is directly obtained by plugging $\mu_{X_i}$ into $x_i$ in Equation 2. Also, when $Cov(x_{i_1}, x_{i_2})$ is the covariance between $x_{i_1}$ and $x_{i_2}$ ($i_1, i_2 \in \{1, 2, \ldots, m\}, i_1 \neq i_2$), the variance of $y_j$ becomes

$$\sigma_{Yj}^2 = V\left[\sum_i w_{i,j}x_i + b_j\right] = V\left[\sum_i w_{i,j}x_i\right] = \sum_i w_{i,j}^2 \sigma_{Xi}^2 + 2\sum_{i_1 < i_2} w_{i_1 j}w_{i_2 j}Cov(x_{i_1}, x_{i_2}).$$
$$\tag{3}$$

In addition, we use $p_{X_i Y_j}(x_i, y_j)$ as the probability of the joint distribution of $x_i$ and $y_j$. When $\rho_{i,j} \in \mathbb{R}$ is the correlation coefficient between $x_i$ and $y_j$, the absolute value of $\rho_{i,j}$ is maximized to 1 when $|w_{i,j}|/\sum_i |w_{i,j}| = 1$ since $y_j = w_{i,j}x_i + b_j$, and minimized to 0 when $|w_{i,j}|/\sum_i |w_{i,j}| = 0$.

Using $\rho_{i,j}$, $p_{XiYj}(x_i, y_j)$ can be written as (Yost, 1984)

$$p_{XiYj}(x_i, y_j) = \frac{1}{2\pi\sigma_{Xi}\sigma_{Yj}\sqrt{1-\rho_{i,j}^2}}$$

$$\exp\left(-\frac{1}{2(1-\rho_{i,j}^2)}\left(\frac{(x_i-\mu_{Xi})^2}{\sigma_{Xi}^2} - 2\rho_{i,j}\frac{(x_i-\mu_{Xi})(y_j-\mu_{Yj})}{\sigma_{Xi}\sigma_{Yj}} + \frac{(y_j-\mu_{Yj})^2}{\sigma_{Yj}^2}\right)\right).$$
(4)

To evaluate the impact of transplantation in the neural network, we measure the representation of the model by the mutual information, which is a Shannon entropy-based measure of dependence between random variables. It is also used to measure the transmission of information between layers (Fan et al., 2021).

In the process of transplantation, the weight of an inactive $j$-th cell $\boldsymbol{w}_j$ is swapped into that of an active $j'$-th cell $\boldsymbol{w}_{j'}$ with the larger variance, where $\sigma_{Yj'}^2 > \sigma_{Yj}^2$. For the mutual information $I(X;Y)$, let $T(I(X;Y))$ be the mutual information after transplant. Since we only change the weights when transplanting, the amount of the change of the mutual information can be described as

$$T(I(X;Y)) = \int_x \int_y \left(p_{XY}(x,y) + \Delta_{\text{tr}}p_{XY}(x,y)\right)\log\frac{p_{XY}(x,y) + \Delta_{\text{tr}}p_{XY}(x,y)}{p_X(x)\left(p_Y(y) + \Delta_{\text{tr}}p_Y(y)\right)}dxdy, \quad (5)$$

where $\Delta_{\text{tr}}$ describes the variation when we apply transplant. Using Equation 2, Equation 4, and Equation 5, we have

$$\Delta_{\text{tr}}p_{XY}(x,y) =$$

$$\frac{1}{mn}\left(\sum_{j'\in S'}\frac{1}{2\pi\sigma_{Xi}\sqrt{\sum_i w_{i,j'}^2\sigma_{Xi}^2 + 2w_{i_1j'}w_{i_2j'}\sum_{i1<i2}Cov(x_{i_1},x_{i_2})}\sqrt{1-\rho_{i,j'}^2}}\right.$$

$$\exp\left(-\frac{1}{2(1-\rho_{i,j'}^2)}\left(\frac{(x_i-\mu_{Xi})^2}{\sigma_{Xi}^2}\right.\right.$$

$$-2\rho_{i,j'}\frac{(x_i-\mu_{Xi})(\sum_i(x_i-\mu_{Xi})w_{i,j'})}{\sigma_{Xi}\sqrt{\sum_i w_{i,j'}^2\sigma_{Xi}^2 + 2w_{i_1j'}w_{i_2j'}\sum_{i1<i2}Cov(x_{i_1},x_{i_2})}}$$

$$\left.\left.+\frac{(\sum_i(x_i-\mu_{Xi})w_{i,j'})^2}{\sum_i w_{i,j'}^2\sigma_{Xi}^2 + 2w_{i_1j'}w_{i_2j'}\sum_{i1<i2}Cov(x_{i_1},x_{i_2})}\right)\right)$$

$$-\sum_{j\in S}\frac{1}{2\pi\sigma_{Xi}\sqrt{\sum_i w_{i,j}^2\sigma_{Xi}^2 + 2w_{i_1j}w_{i_2j}\sum_{i1<i2}Cov(x_{i_1},x_{i_2})}\sqrt{1-\rho_{i,j}^2}}$$

$$\exp\left(-\frac{1}{2(1-\rho_{i,j}^2)}\left(\frac{(x_i-\mu_{Xi})^2}{\sigma_{Xi}^2}\right.\right.$$

$$-2\rho_{i,j}\frac{(x_i-\mu_{Xi})(\sum_i(x_i-\mu_{Xi})w_{i,j})}{\sigma_{Xi}\sqrt{\sum_i w_{i,j}^2 V(x_i) + 2w_{i_1j}w_{i_2j}\sum_{i1<i2}Cov(x_{i_1},x_{i_2})}}$$

$$\left.\left.\left.+\frac{(\sum_i(x_i-\mu_{Xi})w_{i,j})^2}{\sum_i w_{i,j}^2\sigma_{Xi}^2 + 2w_{i_1j}w_{i_2j}\sum_{i1<i2}Cov(x_{i_1},x_{i_2})}\right)\right)\right),$$
(6)

$$\Delta_{\text{tr}}p_Y(y) = \frac{1}{n}\left(\sum_{j'\in S'}\frac{1}{\sqrt{2\pi\sum_i w_{i,j'}^2\sigma_{Xi}^2 + 2w_{i_1j}w_{i_2j'}\sum_{i1<i2}Cov(x_{i_1},x_{i_2})}}\right.$$

$$\exp\left(-\frac{(\sum_i(x_i-\mu_{Xi})w_{i,j'})^2}{2\sum_i w_{i,j'}^2\sigma_{Xi}^2 + 2w_{i_1j}w_{i_2j'}\sum_{i1<i2}Cov(x_{i_1},x_{i_2})}\right)$$

$$-\sum_{j\in S}\frac{1}{\sqrt{2\pi\sum_i w_{i,j}^2\sigma_{Xi}^2 + 2w_{i_1j}w_{i_2j}\sum_{i1<i2}Cov(x_{i_1},x_{i_2})}}$$

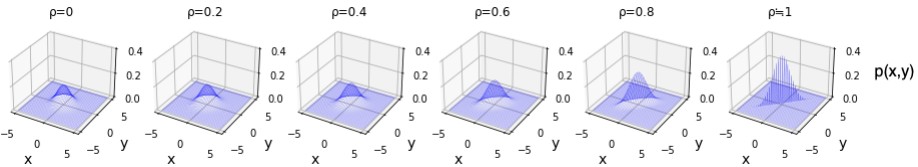

Figure 4: Joint distribution of $x_i$ and $y_j$ with respect to $\rho_{i,j}$

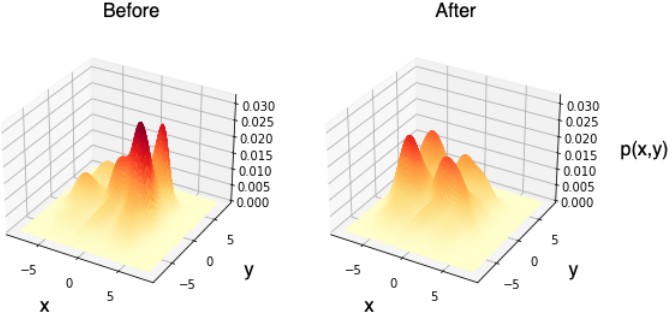

Figure 5: Joint distribution of $x$ and $y$ before/after transplant.

$$\exp\left(-\frac{(\sum_i (x_i - \mu_{Xi})w_{i,j})^2}{2\sum_i w_{i,j}^2 \sigma_{Xi}^2 + 2w_{i_1 j}w_{i_2 j}\sum_{i1<i2} Cov(x_{i_1}, x_{i_2})}\right)\right), \quad (7)$$

where $S'$ is the set of the top $\eta\%$ active cell indices and $S$ is that of the bottom $\eta\%$ inactive cell indices. Furthermore, from the linear connection of $x$ and $y$ in Equation 2, we can understand that $x_i$ and $y_j$ are fully dependent when $|w_{i,j}|/\sum_i |w_{i,j}| = 1$, where $|\rho_{i,j}| = 1$ and $p_{X_iY_j}(x_i, y_j) = p_{Xi}(x_i) = p_{Yj}(y_j) = \sqrt{p_{Xi}(x_i)p_{Yj}(y_j)}$, and $x_i$ and $y_j$ are independent when $|w_{i,j}|/\sum_i |w_{i,j}| = 0$, where $|\rho_{i,j}| = 0$ and $p_{X_iY_j}(x_i, y_j) = p_{Xi}(x_i)p_{Yj}(y_j)$ since the effect of the $i$-th cell $(\sigma_{Xi}w_{i,j})^2$ on the variance of the $j$-th cell $\sigma_{Yj}^2 = \sum_i (\sigma_{Xi}w_{i,j})^2$ changes with the absolute value of $w_{i,j}$. Figure 4 shows the summary for the example of joint distribution when $\rho_{i,j}$ changes. By arranging $w_{i,j}$ based on the balance of $\sigma_{Xi}$ and $\mu_{Xi}$, we can increase $I(X;Y)$. Figure 5 shows an example of the surface of $p_{XY}(x, y)$ before and after transplantation, with parameters $m = 5$, $n = 4$, and $\eta = 25\%$. We can see that the distribution of $p_{XY}(x, y)$ is smoothed by the transplant operation.

Next we discuss the whole impact of the combination of transplant and optimization of neural networks. When we train a neural network, an optimizer continuously updates the weights of the perceptron. Let $f$ be a function that updates weights $\mathbf{w}$ at a given step as

$$f(\mathbf{w}) = \mathbf{w} - g(\nabla \mathcal{L}(\mathbf{w})), \quad (8)$$

where $\mathcal{L}(\mathbf{w})$ is the loss determined by the whole weights and $g(\nabla \mathcal{L}(\mathbf{w}))$ is the update of the weights based on the gradient of the loss, to minimize the loss of the network for each step.

In general, the more training steps $k$; that is, the more $f$ is applied to $\mathbf{w}$, the larger the exploration space of weights. Let $O(I(X;Y))$ be the mutual information after optimization with $k$-step training, between input and output of the perceptron layer which we apply transplant. When we train the model, both weights and the probability distributions of $x$ and $y$ are updated. When $\Delta_{\text{opt}}$ denotes the variation of the probability when we apply a training of $k$ steps, the mutual information $O(I(X;Y))$ after the training can be obtained as Equation 5 by replacing $\Delta_{\text{tr}}$ with $\Delta_{\text{opt}}$.

Therefore, the mutual information $I(X;Y)$ changes into $I(X;Y)'$ such that

$$I(X;Y)' = O \circ (T \circ O)^{(c-1)}(I(X;Y)). \quad (9)$$

after $c \in \mathbb{N}$ times training of $k$ steps.

In the transplant operation, we preferentially adopt weights with the larger variance $\sigma_{Yj}^2$. When the perceptron tries to improve the network by minimizing the loss, well trained cells are expected to

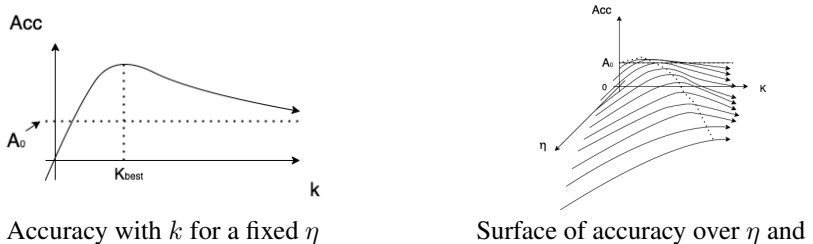

| Accuracy with $k$ for a fixed $\eta$ | Surface of accuracy over $\eta$ and $k$ |

Figure 6: Schematic estimation of behavior of perceptron with transplant.

return balanced, and high variance output. Moreover, since the exploration space of each weight is limited in the $k$-step training, when the larger number of weights are well updated with larger $|w_{i,j}|$, the more the variance $\sigma_{Yj}^2 = \sum_i (\sigma_{Xi} w_{i,j})^2$ becomes. This means that weight selection for larger $\sigma_{Yj}^2$ tends to lead to balanced weights, where $|w_{i,j}| / \sum_i |w_{i,j}|$ is expected to be smoother than the case that the less number of weights are well updated. Therefore, transplant can be considered as stochastic regularization of a neural network. Since an optimizer has to coordinate the weights of the following layers to propagate a varied distribution, we can assume that $k$ has to be large enough to balance the weights after transplantation. In contrast, when $k$ is too large, the performance of the model converges to the state without transplantation, hence transplantation has less impact on the overall training of the model. Figure 6 shows an estimate of the performance behavior of the model.

The changes of the performance with respect to $k$ is considered to be a convex upward function, which is maximized at a certain $k = K_{\text{best}}$, and converges to certain accuracy that coincides with the performance without transplant as $k$ increases. Moreover, when we increase the ratio $\eta\%$ of cells to be transplanted, the optimizer needs more $k$ to arrange the weights, and $K_{\text{best}}$ is considered to become larger. Furthermore, the performance gets maximized with an appropriate $\eta$ to replace redundant cells, while too large $\eta$ will make the accuracy worse, since the transplant will start to replace even active cells. Therefore, the best $\eta$ to maximize the accuracy is also thought to be a convex upward like the right surface in Figure 6.

## 4 EXPERIMENTS

To grasp the behavior of neural networks when we apply transplantation during training, and to validate the activeness we proposed, we empirically investigate transplantation on real-world datasets.

In our experiments, we use the following setup: (1) Report the accuracy for transplantation over parameters $(k, \eta)$ on grid search, and evaluate the mutual information. (2) Compare performance of the model trained with transplant with our proposed activeness and that trained with transplant without using the activeness. (3) Evaluate the performance of models with different architectures, and test the distributions of inputs and outputs for the middle layer. (4) Test the effect of the transplant on different datasets. For all experiments, we use Ubuntu Linux (version: 4.15.0-117-generic) and run all experiments on 2.20 GHz Intel Xeon E5-2698 CPU with 252 GB of memory, and Tesla V100 GPU with 32GB of memory.

### 4.1 RESULTS OF TRANSPLANT

To evaluate the effect of transplantation, we use the MNIST dataset (LeCun et al., 1998), which consists of a training set of 60,000 instances and a test set of 10,000 instances. Each instance is a 28x28 grayscale image associated with a label from 10 classes of digits. In this experiment, the network is a simple architecture with the 2 layer perceptron, which contains a fully connected layer with 100 cells as the target of transplantation, and the other classification layer. We use a learning rate of 0.0003 and train the network for 20 epochs with a batch size of $\beta = 10$. We transplant $\eta\%$ of the cells in the target layer for the checkpoint after every $k$ iterations, and evaluate the accuracy of training and validation. To confirm the behavior of the score when the parameters change gradually, we experiment the accuracy of the model for all combinations of $k$ in $100, 200, 500, 1000, 1500, 2000, 2500$, and $\eta$ in $0, 1, 2, 3, 4, 5, 6, 7, 8, 9$.

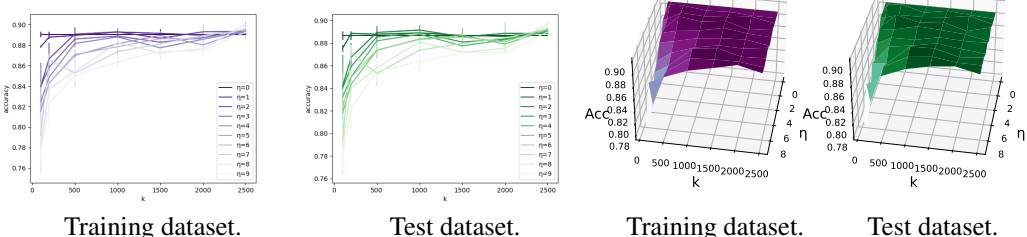

| Training dataset. | Test dataset. | Training dataset. | Test dataset. |

Figure 7: Accuracy under various $k$ and $\eta$ on the MNIST dataset.

Table 1: Mutual information with/without transplant on MNIST dataset.

| Method | Mutual Information |
|---|---|
| Without transplant | $0.0002058 \pm 0.0000117$ |
| Transplant | $0.0004514 \pm 0.0000392$ |

The summary of the results is shown in Figure 7, where the accuracy curve with respect to changes of $k$ is roughly convex upwards. The surface of the accuracy forms a hilly curve over $k$ and $\eta$, and is maximized at certain values as expected from our theoretical discussion. Here we used neither activation layers nor a large number of cells to directly validate our theoretical analysis. Thus the accuracy obtained in our experiments is lower than that of the state-of-the-art MLP models.

After training each model, we evaluate the mutual information between the input layer and the hidden layer, for the model trained without transplantation, and the model trained with transplantation of the best parameters $k$ and $\eta$. Results are shown in Table 1. Figure 8 also compares probability distributions of $y$ in the original training without transplant and that with transplant. The parameters $k$ and $\eta$ for the transplant are set to the best values in Experiment 4.1. As we expected, the distribution with transplant is smoother, and the variance of $y$ is larger.

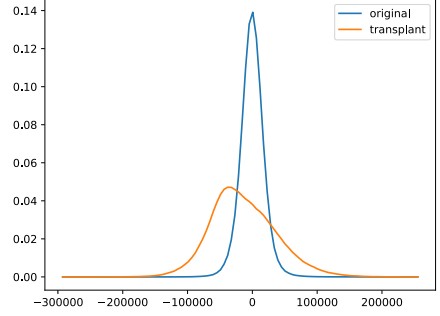

Figure 8: Output distributions with/without transplant.

To evaluate the effectiveness of the activeness we have proposed, we compare the performance via the transplant under the proposed activeness and transplant that randomly switches the weights without using the activeness. We run the experiment with the same parameters of $(k, \eta)$ as in the previous experiment in Section 4.1 and show results in Figure 9. We can see that the resulting accuracy with the activeness behaves significantly more convex than the random switching, and gets better accuracy overall.

## 4.2 RESULTS IN MULTIPLE ARCHITECTURES

Since transplantation can be applied to any architectures, we consider different model architectures by increasing the number of layers in the MLP and evaluate the effect of transplantation. Each model is trained with the same parameters as in Section 4.1, while we apply transplant to all hidden layers. We train the 3-layer MLP, which has 2 hidden layers to transplant, and the 4-layer model, which has 3 hidden layers to transplant. Results are summarized in Table 2. We can confirm the improvement in accuracy due to the transplantation for all architectures.

After training of the perceptron with 3 layers with the best parameters of $(\eta, k)$, we perform prediction on all the test data with the model and plot the joint distribution between input $x$ and output $y$ of the model with/without transplantation in Figure 10. We can see that the distribution becomes smoother when we apply transplantation during training.

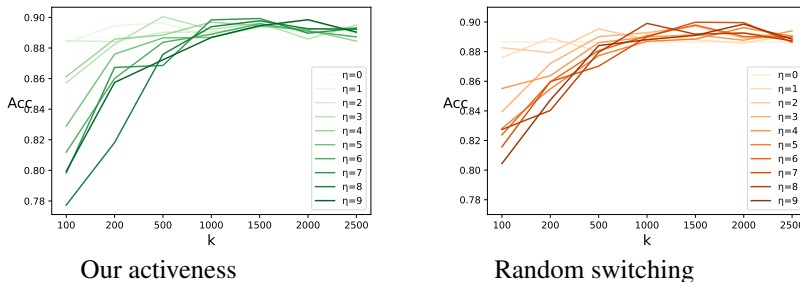

Figure 9: Comparison of transplant based on our activeness and random switching.

Table 2: Results of Experiment 4.3

| Number of layers | Accuracy without transplant | Accuracy with transplant |
|---|---|---|
| 1 | 0.8870 | 0.9005 |
| 2 | 0.8812 | 0.8945 |
| 3 | 0.8819 | 0.8930 |

Also, we show the scatter plots of some samples of $(x_i, y_j)$ in Figure 12 of Appendix 1. We can see that transplant arranges the weights of the model as they increase the mutual information between $X$ and $Y$ by increasing the correlation $\rho_{i,j}$ as shown in Figure 4.

### 4.3 RESULTS ON DIFFERENT DATASETS

We also examine the behavior of the model trained with transplantation on different datasets. We follow the same protocol of our experiment in Section 4.1 on three other classification benchmarks. (1) Fashion MNIST (Xiao et al., 2017), which consists of a training set of 60,000 instances and a test set of 10,000 instances. Each instance is a 28x28 grayscale image associated with a label from 10 classes of fashion items. We use a learning rate of 0.0003 and train the network for 20 epochs with a batch size of $\beta = 10$. We searched for $k$ in $100, 200, 500, 1000, 1500, 2000, 2500$, and $\eta$ in $0, 1, 2, 3, 4, 5, 6, 7, 8, 9$. (2) CIFAR-10 (Krizhevsky, 2009), which consists of a training set of 50,000 instances and a test set of 10,000 instances. Each instance is a 28x28 color image, associated with a label from 10 classes. We use a learning rate of 0.0003 and train the network for 20 epochs with a batch size of $\beta = 10$. We searched for $k$ in $148, 181, 221, 270, 330, 403, 492, 601, 735, 897, 1096$, and $\eta$ in $0, 1, 2, 3, 4, 5, 6, 7, 8, 9$. (3) Mushroom dataset (Lincoff, 1981), consisting of 8124 instances. We randomly split $80\%$ of them as a training set and $20\%$ of them as a test set. Each example has 12 dimensional features about mushrooms, associated with a binary class of edibility. We use a learning rate of 0.0003 and train the network for 50 epochs with a batch size of $\beta = 10$. We searched for $k$ in $148, 181, 221, 270, 330, 403, 492, 601, 735, 897, 1096$, and $\eta$ in $0, 1, 2, 3, 4, 5, 6, 7, 8, 9$.

Results are shown in Figure 11. In the MNIST and Fashion MNIST datasets, we can see the clear relationship of the convex function between $k$ and the accuracy, according to the value of $\eta$. Also, the relationship can be observed in the Mushroom dataset, but in the CIFAR-10 dataset, the effect of transplantation is relatively difficult to find, because the value of accuracy is too low, and varies widely throughout results. From these experiments, we can assume that transplantation generally improves the performance of the model by regularizing the weights, regardless of the task.

## 5 RELATED WORK

There is a research field called "grow-and-prune" (Lemeng et al., 2020; Sokar et al., 2023; Xiaoliang et al., 2019), which initializes a part of the model while training networks, inspired by the biological brain function "synaptic pruning", in which excess neural connections exist in the brains of newborn animals, but eventually the necessary connections are strengthened and the unnecessary ones are removed, and the neural circuit matures. The idea of transplant is related to the genetic algo-

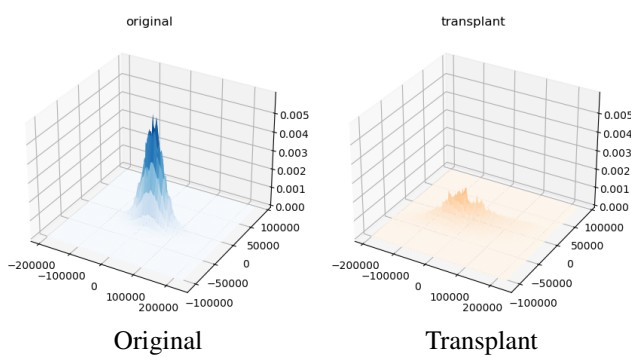

Original                                    Transplant

Figure 10: Joint distribution of input and output of 3-layer MLP with/without transplant.

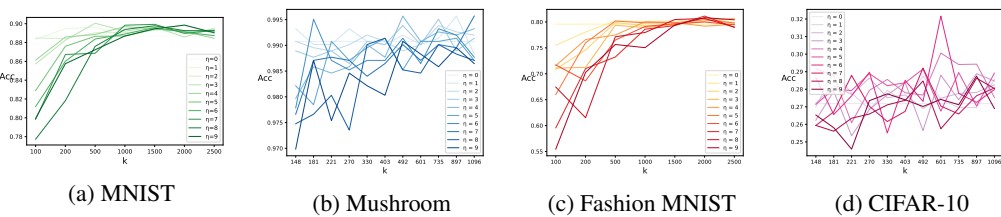

(a) MNIST                    (b) Mushroom            (c) Fashion MNIST            (d) CIFAR-10

Figure 11: Accuracy under various $k$ and $\eta$ on four datasets.

rithm (Sastry et al., 2005; Katoch et al., 2021), which is inspired by the phenomenon that "stronger individuals that adapt to their environment survive, while weaker individuals that cannot adapt to their environment are weeded out", which occurs in the process of biological evolution. Genetic algorithms are a mechanism for passing on superior individuals to the next generation in a programmed manner. There are some discoveries by applying genetic algorithms to neural networks. For example, the idea of dropout (Srivastava et al., 2014) was motivated from a theory of the role of sex in evolution (Livnat et al., 2010), and it improves the robustness of the model. The genetic algorithm is also classified as a type of the evolutionary algorithm (Cheng et al., 2016), which is a population-based metaheuristic optimization algorithm. The evolutionary algorithm uses algorithms inspired by evolutionary mechanisms such as reproduction, mutation, genetic modification, natural selection, and survival of the fittest as its mechanism. It is also proposed to improve optimization methods including neural networks with the natural selection of evolutionary algorithms (Vrugt & Robinson, 2007; Mirjalili, 2019), or updating whole weights of the model with the mutation and the crossover, instead of using backpropagation (Montana & Davis, 1989). However, these genetic crossovers occur only in the alternation of generations in the process of biological evolution. In contrast, we apply the transplantation of weights periodically and alternately after some steps of the training with backpropagation, as organisms acquire the ability during lifetime and leave a legacy to the next generation.

## 6    CONCLUSION

We have proposed the concept to "transplant" cells in the perceptron, as neuronal cells in the brain are replaced for the purpose of therapy in the field of neurobiology. We have theoretically analyzed how the performance of the model behaves when we apply transplantation. We have also obtained the experimental feedback to support the theoretical analysis. Finally, we have shown that the idea of "transplant", which is cybernetically inspired, can improve the neural networks that contain at least one perceptron layer.

**Ethics statement:** We do not have any ethics issues.

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

# A APPENDIX

## A.1 CHANGE OF EACH CELL'S INPUT AND OUTPUT

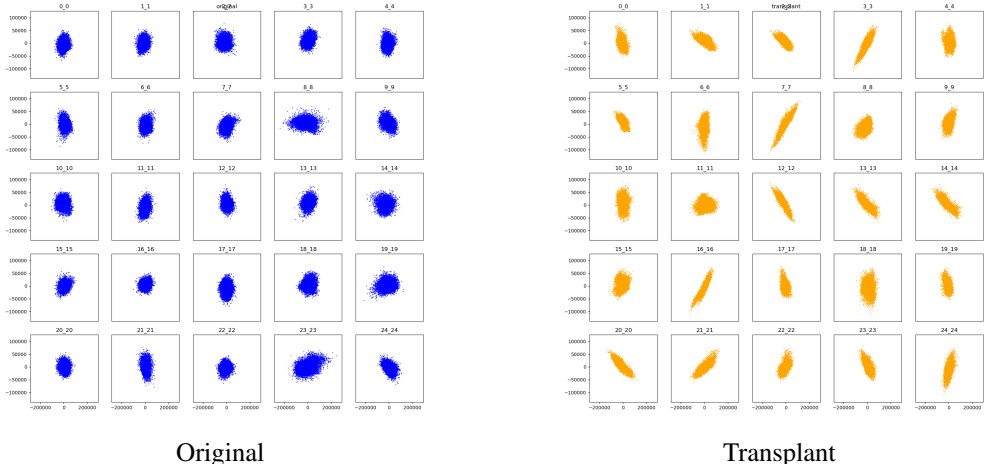

Original          Transplant

Figure 12: Scatter plots of input and output of each cell of 3-layer MLP with/without transplant.

