# OpenReview forum: "Transplant of Perceptrons"
_ICLR.cc/2024/Conference — Submitted to ICLR 2024_

### Official Review · Reviewer_azj9 · 2023-10-29

**Soundness:** 2 fair
**Presentation:** 2 fair
**Contribution:** 2 fair
**Rating:** 3
**Confidence:** 4

**Summary:**

The paper aims to simulate the neurobiological phenomenon of transplantation and proposes an approach for implementing it in MLPs. It utilizes neuron variance as a measure of activity and replaces non-active neurons with active ones. The paper also offers theoretical insights into how this transplantation method enhances mutual information. Finally, empirical experiments are primarily conducted using MNIST.

**Strengths:**

- The paper highlights the neurobiological phenomenon of transplantation, which is quite interesting.

- An attempt is made to provide theoretical insights into the benefits of their approach.

**Weaknesses:**

- While the idea is novel and interesting, the major weakness of the paper lies in its implementation, applicability, and lack of empirical validation.
- The authors chose variance as the criterion for activeness and drew parallels with Hebbian learning. However, the chosen approach fails to take into account the class label, as it is expected that some neurons would specialize for a certain class. The proposed approach might also hinder the specialization of neurons, which is a desirable trait.
- The theoretical analysis needs to be more concise and clearly state actionable insights. The chosen approach makes it seem like an expected result that mutual information would increase.
- I am not convinced that simply copying the weights of the active neurons to inactive neurons is an optimal approach for implementing transplantation. The authors also fail to provide a convincing justification for this choice.
- The biggest concern lies with the empirical validation and the applicability of the method. The primary analysis is conducted on MNIST, which is not an issue in itself. However, the baselines are much lower than those of traditional MLPs. The decision not to utilize activation layers is concerning and lacks justification. This raises questions about whether their findings hold with activation layers, which are essential for the applicability of the method.
- While the authors claim that their method is applicable to different architectures, they have not demonstrated this. They only considered different numbers of hidden layers in MLPs, which does not truly constitute a different architecture. The applicability of their approach to CNNs, for instance, remains unclear. Furthermore, it's important to consider how activation layers and batch normalization would affect their findings.
- Extending the study to more complex datasets like CIFAR-10 is also left unclear.

**Questions:**

Please refer to the weaknesses. In general, the authors should provide clearer justifications for their design choices and address the shortcomings and concerns in their empirical evaluation.

---

### Official Review · Reviewer_wDLW · 2023-11-01

**Soundness:** 2 fair
**Presentation:** 2 fair
**Contribution:** 2 fair
**Rating:** 3
**Confidence:** 3

**Summary:**

The authors propose a method to modify trained NNs by replacing the weights of units with very low activity by the weights of units with highest activity. They argue that according to a metric of mutual information the NN is thus improved. The results indicate very similar effects to randomly reassigning the weights feeding into low activity units.

**Strengths:**

I appreciate the boldness of the program and the writing.

Also, the bibliography was the only one of all the papers I reviewed for ICLR that was formatted for easy searching.

**Weaknesses:**

The connection to forms of biological neuronal replacement are overstated. In particular, even if neurons could be replaced 1 for 1 in a brain, the goal is to replace a previously valuable but damaged neuron. This is a different program than the paper pursues in ANNs.

To reset the weights feeding into or out of units to high values, after those weights have been trained into irrelevance, is highly counterintuitive, and thus requires a very strong argument to justify. In this case, a non-trivial increase in accuracy would be needed (this does not happen).

The results are for NNs with linear activation functions. How the proposed method would work in NNs with non-linear activation functions is important but not addressed.

Reviewer limitation: I did not read section 3 (Theoretical Analysis).

**Questions:**

General review context: Please note that I am simply another researcher, with some overlap of expertise with the content of the paper. In some cases my comments may reflect points that I believe are incorrect or incomplete. In most cases, my comments reflect spots where an average well-intentioned reader might stumble for various reasons, reducing the potential impact of the paper. The issue may be the text, or my finite understanding, or a combination. These comments point to opportunities to clarify and smooth the text to better convey the intended story. I urge the authors to decide how or whether to address these comments. I regret that the tone can come out negative even for a paper I admire; it's a time-saving mechanism for which I apologize.

Thanks for a clearly formatted bibliography, which made checking up on references easy.

Abstract: "transplant" in the context of neuroscience, to the extent that I am aware, does not parallel the transplanting proposed in this paper, so the BNN-ANN parallel is not motivating.

"such an interaction has decreased": I don't think this is true. I believe it is an active and fertile area of research. Some years ago I wrote multiple papers in this area.

"is hardly limited": meaning is unclear

Top of pg 3: For Hebbian learning, both neurons must be active at the same time. How does the definition of "activity" (previous page) for ANN units align with this? It sounds like activity is the variance of one unit's output.

top of pg 6 "rutern balanced and high variance output": Is this true? I am not familiar with this.

Fig 7 formatting: too small for me. Axis labels and tick labels are too small. Perhaps give more detail in the caption to orient the reader to the important points in the images.

Fig 9: I do not see a difference in A and B. Also, what does the unchanged trained network produce (as a baseline)? I ask because the transplanting seems to be undoing the effects of training.

Table 2: Can this include +/- std dev over multiple runs? This makes clear whether the difference between the two methods is within a noise envelope (see also the text re results for CIFAR-10 near the end of section 4).

---

### Official Review · Reviewer_RbUD · 2023-11-01

**Soundness:** 1 poor
**Presentation:** 1 poor
**Contribution:** 1 poor
**Rating:** 1
**Confidence:** 5

**Summary:**

Inspired by neuroscience studies demonstrating the effectiveness of neuronal cell transplantation, the paper presents a method for transplantation in the perceptron model. The paper achieves transplantation in the perceptron by first measuring the variance of a neuron/cell across $k$ batches of data. The variance will serve as the activeness of the cell, inspired by Hebbian theory.  Transplantation in the perceptron will effectively duplicate the activity of high-variance neurons into lower-variance neurons.

The authors empirically compare the effectiveness of their transplantation method against having no transplantation and having random transplantation. They state that their transplantation method is more effective than random transplantation and no transplantation.

**Strengths:**

The idea of a "transplant" is an original and interesting approach for Hebbian-like update rules in multi-layer perceptrons. Artificial neural networks could certainly benefit from research in cell therapy.

**Weaknesses:**

My biggest concern in this paper is the performance of the transplanted multi-layer perceptron: in the ablation results for $\eta$ and $k$, the accuracy does not surpass $90$% for MNIST, including $\eta=0$ (vanilla multi-layer perceptron). It is widely accepted that MNIST achieves an accuracy of $>90$% for MNIST on logistic regression. Hence, the multi-layer perceptron described in the paper (with a hidden layer of width 100) should comfortably achieve $>90$% for MNIST.

My second biggest concern is the strong abuse of the term perceptron. The perceptron refers to a single-neuron such that $y=wx + b$, where $w$ is a **vector** of real-valued weights. In the paper, the authors define a perceptron as having a weight matrix $W \in \mathbb{R}^{m \times n}$, which is rather a multi-layer perceptron. These are fundamental concepts that should not be abused. In particular, Hebbian learning in a perceptron is very well understood, as having the update rule $w_{t+1} = w_t + yx$, it would have been advantageous to highlight this to describe hebbian learning in perceptrons. However, I believe in the manuscript, the authors were largely referring to multi-layer perceptrons and not the perceptron.

Generally, I don't find that the paper highlights or addresses a problem with artificial neural networks beyond seeking inspiration from neurobiology. Why would we need transplantation in neural networks?

I found the mutual information analysis lacks clarity: the authors compute the mutual information between the data $x$ and the output $y$ and compare the different transplantation methods. However, I don't find that the authors sufficiently motivate why this analysis is important and how this relates to transplantation.

Minor feedback:

- Section 2.2 is not necessary, and can be moved to the appendix.
- Figures 4, 7, 9, and 11 are very small, it would be good to make them larger to make them more legible.
- The algorithm is relatively simple, would be nice to make the code available.

**Questions:**

In Figure 5, the authors claim that transplantation "smoothens" the joint distribution of $x$ and $y$. Could you elaborate on what does smoothness mean in this case?

---

### Official Review · Reviewer_Vew5 · 2023-11-05

**Soundness:** 3 good
**Presentation:** 3 good
**Contribution:** 2 fair
**Rating:** 5
**Confidence:** 3

**Summary:**

The paper presents a biologically-inspired method to improve DNN training by replacing inactive neurons (where activeness is measured as the variance of the output) by copying over weights and biases of more active ones to the inactive neurons during training. The validity of the approach is evaluated on MNIST dataset.

**Strengths:**

The paper is clearly written and illustrations support the text well. The method is generic and seems to be easy to implement, so there is a potential for high impact.

**Weaknesses:**

1) It would be great to see a more in-depth analysis of Fig. 8: the proposed method seems to degrade the quality; how is this justified?

2) Sec. 4.2 is titled 'RESULTS IN MULTIPLE ARCHITECTURES' which for a conventional reader is a bit misleading - the architectures evaluated are 1-4 layer perceptrons.

3) It would be great to see the application of the approach on a more modern task/DNN - e.g. sota image segmentation or 3D reconstruction / egomotion estimation. Given the relative simplicity of the method, what would be the technical challenges implementing such evaluation?

**Questions:**

Fig. 9 - it is quite difficult for the reader to compare left and right images; it might be easier to look at the difference between the plots.

---

### Meta-Review · Area_Chair_aT4b · 2023-12-05

**Metareview:**

The paper draws inspiration from neuroscientific findings on neural cell transplantation and proposes to "copy" useful weights/neurons in replacement of inactive/dead ones in neural architectures. Whereas reviewers agree on the novelty on the idea, there are several major concerns on the use and description of terminology, as well as on the experimental execution of the paper. Overall, the applicability in the present form is limited, and more evidence is required in broader set-ups and tasks.

**Justification For Why Not Higher Score:**

The reviewers are all in agreement on the points that require improvement in the present version of the paper. Primarily, this includes disambiguation and misuse of terminology, more exhaustive set-ups and extensive evaluation. During the discussion phase, no rebuttal or responses to the reviewer feedback were submitted.

**Justification For Why Not Lower Score:**

N/A

---

### Decision · Program_Chairs · 2024-01-16

Reject